# Experience shapes activity dynamics and stimulus coding of VIP inhibitory cells

Marina Garrett*, Sahar Manavi, Kate Roll, Douglas R Ollerenshaw, Peter A Groblewski, Nicholas D Ponvert, Justin T Kiggins, Linzy Casal, Kyla Mace, Ali Williford, Arielle Leon, Xiaoxuan Jia, Peter Ledochowitsch, Michael A Buice, Wayne Wakeman, Stefan Mihalas, Shawn R Olsen*

Allen Institute for Brain Science, Seattle, United States

**Abstract** Cortical circuits can flexibly change with experience and learning, but the effects on specific cell types, including distinct inhibitory types, are not well understood. Here we investigated how excitatory and VIP inhibitory cells in layer 2/3 of mouse visual cortex were impacted by visual experience in the context of a behavioral task. Mice learned a visual change detection task with a set of eight natural scene images. Subsequently, during 2-photon imaging experiments, mice performed the task with these familiar images and three sets of novel images. Strikingly, the temporal dynamics of VIP activity differed markedly between novel and familiar images: VIP cells were stimulus-driven by novel images but were suppressed by familiar stimuli and showed ramping activity when expected stimuli were omitted from a temporally predictable sequence. This prominent change in VIP activity suggests that these cells may adopt different modes of processing under novel versus familiar conditions.

## Introduction

Neural circuits are dynamically shaped by experience and learned expectations (*de Lange et al., 2018*; *LeMessurier and Feldman, 2018*; *Pakan et al., 2018*; *Ranganath and Rainer, 2003*). Visual experience can modify cortical representations, including changes in gain, selectivity, correlations, and population dynamics (*Jurjut et al., 2017*; *Khan et al., 2018*; *Makino and Komiyama, 2015*; *Poort et al., 2015*; *Weskelblatt and Niell, 2019*; *Woloszyn and Sheinberg, 2012*). Moreover, sensory and behavioral experience can lead to the emergence of predictive activity in the visual cortex including reward anticipation (*Poort et al., 2015*; *Shuler and Bear, 2006*), spatial expectation (*Fiser et al., 2016*; *Saleem et al., 2018*), anticipatory recall (*Gavornik and Bear, 2014*; *Xu et al., 2012*) and prediction error signals (*Fiser et al., 2016*; *Hamm and Yuste, 2016*; *Homann et al., 2017*). These learning-related changes in sensory cortex can involve top-down feedback (*Fiser et al., 2016*; *Makino and Komiyama, 2015*; *Petro et al., 2014*; *Zhang et al., 2014*) and neuromodulatory inputs (*Chubykin et al., 2013*; *Kuchibhotla et al., 2017*; *Pinto et al., 2013*), and may be associated with a shift in the balance of bottom-up sensory and top-down contextual signals (*Batista-Brito et al., 2018*; *Khan and Hofer, 2018*). Inhibitory interneurons likely play a key role in this process by dynamically regulating the flow of information (*Hangya et al., 2014*; *Kepecs and Fishell, 2014*; *Wang and Yang, 2018*). Elucidating how different cell populations, particularly inhibitory cells, contribute to experience-dependent changes in sensory coding is critical to understand the dynamic nature of cortical circuits.

Vasoactive intestinal peptide (VIP) expressing cells comprise a major class of inhibitory neurons and are well-positioned to mediate top-down and neuromodulatory influences on local circuits in sensory cortex. VIP cells receive long-range projections from frontal areas (*Lee et al., 2013*; *Wall et al., 2016*; *Zhang et al., 2016*; *Zhang et al., 2014*) as well as cholinergic and noradrenergic inputs (*Alitto and Dan, 2013*; *Fu et al., 2014*). VIP cells are highly active during states of arousal

*For correspondence:
marinag@alleninstitute.org (MG);
shawno@alleninstitute.org (SRO)

Competing interests: The authors declare that no competing interests exist.

(*Fu et al., 2014*; *Reimer et al., 2014*), are modulated by task engagement (*Kuchibhotla et al., 2017*), and are responsive to behavioral reinforcement (*Krabbe et al., 2019*; *Letzkus et al., 2011*; *Pi et al., 2013*). In the local cortical circuitry, VIP cells primarily inhibit another major class of inhibitory interneuron, somatostatin (SST) cells (*Lee et al., 2013*; *Munoz et al., 2017*; *Pfeffer et al., 2013*; *Pi et al., 2013*), which can result in disinhibition of excitatory neurons (*Fu et al., 2017*; *Lee et al., 2013*; *Letzkus et al., 2011*). SST cells target the apical dendrites of pyramidal neurons (*Kepecs and Fishell, 2014*) and removal of this inhibition may facilitate the association of top-down and bottom-up input by pyramidal cells (*Chen et al., 2015*; *Larkum, 2013*; *Makino and Komiyama, 2015*). However, little is known about how VIP cell activity is modified by visual experience.

Here we investigated how long-term behavioral experience with natural scene images alters activity of cortical VIP inhibitory and excitatory pyramidal cells in layers 2/3 of mouse visual cortex. Mice were trained to perform a change detection task in which images were presented in a periodic manner and mice were rewarded for detecting changes in image identity. Mice learned the task with one set of eight natural images, which were viewed thousands of times and were thus highly familiar. During subsequent 2-photon imaging, these familiar images as well as three novel image sets were tested. Familiar images were associated with lower overall population activity in both excitatory and VIP cells. Notably, VIP inhibitory cells had distinct activity dynamics during sessions with familiar versus novel images. VIP cells were stimulus-driven by novel images but displayed ramping activity between presentations of familiar images and were suppressed by stimulus onset. These cells showed even greater ramping activity when an expected stimulus was omitted from the regular image sequence. Overall, these results show distinct experience-dependent changes in two cortical cell classes and suggest that VIP cells may adopt different modes of processing during familiar versus novel conditions.

## Results

### Visual change detection task with familiar and novel images

We trained mice on a go/no-go visual change detection task with natural scene stimuli. In this task, mice see a continuous stream of repeatedly presented images (250 ms stimulus presentation followed by 500 ms gray screen; *Figure 1A,B*). On 'go' trials, the image identity changes and mice report the change by licking a reward spout within 750 ms (*Figure 1B,C*). False alarms are quantified during 'catch' trials when the image does not change. To test whether expectation signals exist in the visual cortex due to the temporal regularity of this task, we randomly omitted ~5% of all image presentations (not including image changes to avoid interfering with behavior performance). These omissions appeared as an extended gray period to the mouse and corresponded to a gap in the periodic timing of stimuli (*Figure 1D*).

Mice learned the task through a series training stages, starting with oriented gratings and then progressing to natural images (*Figure 1E–G*; see Materials and methods for additional details about training procedure). During the natural image stage, mice were trained with one set of eight images (image set A) for an extended number of sessions (range = 6–46 sessions with image set A, median = 17 sessions; *Figure 1H*, *Figure 1—figure supplement 1A*). On average, mice viewed each of the eight images from the familiar set 10,350 times prior to the 2-photon imaging stage (range: 944–26,784 individual stimulus presentations per image).

During the 2-photon imaging portion of the experiment, mice performed the task with either the familiar image set or one of three additional novel image sets (*Figure 1F*). Hit rates, false alarm rates, and reaction times (lick latency) were similar across image sets (*Figure 1I,J*). During the task, mice are free to run on a circular disk and typically stop running to lick. There was no difference in running behavior between novel and familiar image sessions (*Figure 1—figure supplement 1B,C*). Licking behavior around the time of omission was also similar across image sets (*Figure 1—figure supplement 1E,F*). Together these results show that mouse behavior was similar for novel and familiar images.

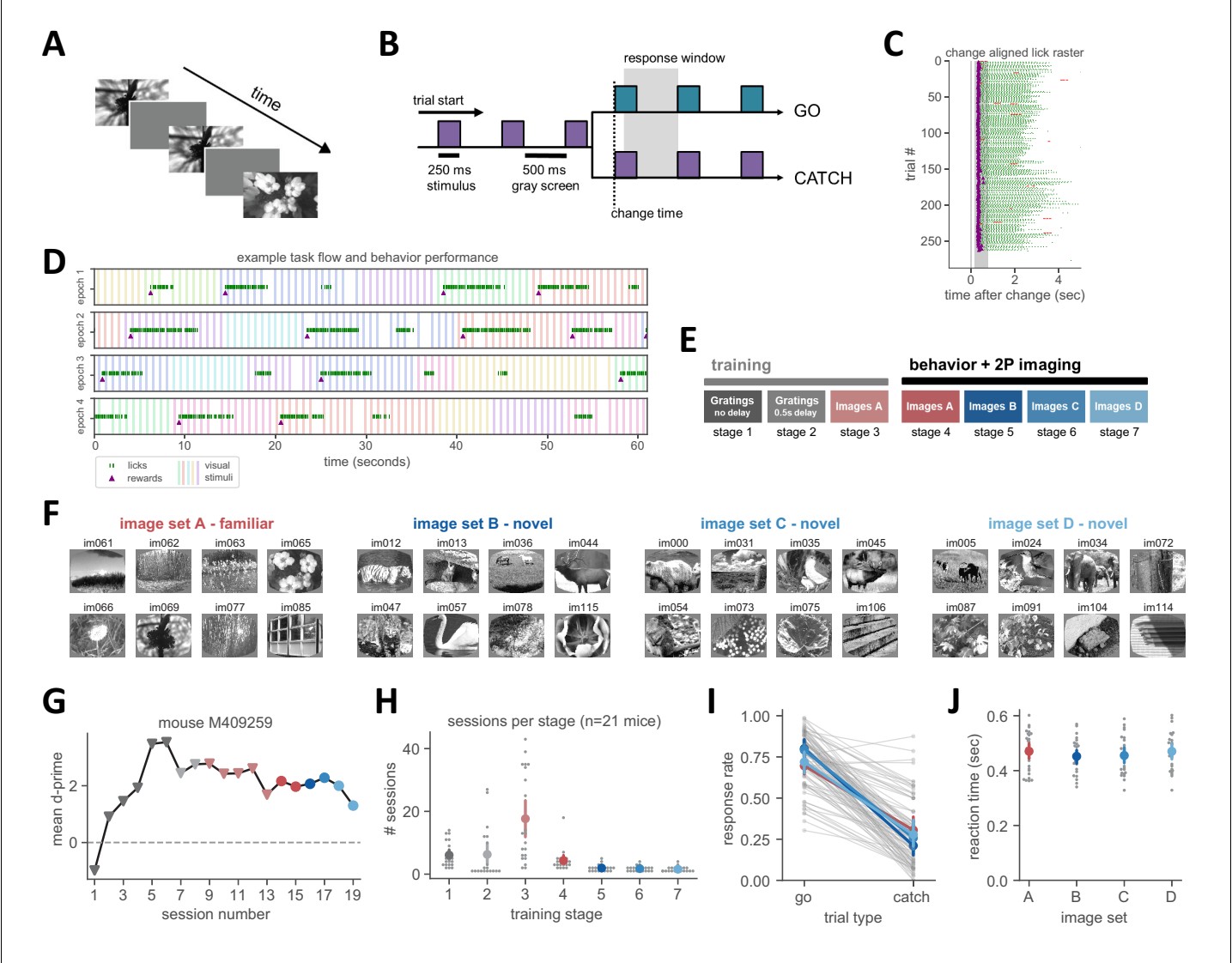

**Figure 1.** Natural image change detection task with familiar and novel images. (**A**) Schematic of stimulus presentation during the task. Images are presented for 250 ms followed by 500 ms of gray screen. (**B**) Trial structure. Colors represent different images. On go trials, the image identity changes and mice must lick within the 750 ms response window to receive a water reward. On catch trials no image change occurs and the behavioral response is measured to quantify guessing behavior. (**C**) Example lick raster, aligned to the image change time. Purple dots indicate rewards and green ticks are reward consumption licks. Red ticks indicate incorrect licking responses outside the response window. (**D**) Example behavior performance over four minutes of one session, separated into one-minute epochs. Colored vertical bars indicate stimulus presentations (different colors are different images). Green tick marks indicate licks, purple triangles indicate rewards. 5% of all non-change image flashes are omitted, visible as a gap in the otherwise regular stimulus sequence. (**E**) Training stages. Mice are initially trained with gratings of 2 orientations, first with no intervening gray screen (stage 1), then with a 500 ms inter-stimulus delay (stage 2). Next, mice perform change detection with eight natural scene images (stage 3, image set A). During the 2-photon imaging portion of the experiment, mice are tested with image set A as well as three novel image sets (B, C, D) on subsequent days. (**F**) The four sets of 8 natural images. Image set A is the familiar training set, and image sets B, C and D were the novel sets shown for the first time during 2-photon imaging. (**G**) Example training time course of one mouse. (**H**) Number of sessions spent in each stage across mice. Mean ± 95% confidence intervals in color, individual mice in gray. (**I**) Response rates for go and catch trials are similar across image sets. Individual behavior sessions are shown in gray and average ± 95% confidence intervals across sessions for each image set are shown in color. (**J**) Reaction times, measured as latency to first lick, are not significantly different across image sets. Mean ± 95% confidence intervals in color, individual sessions in gray.

The online version of this article includes the following figure supplement(s) for figure 1:

**Figure supplement 1.** Behavior is similar across image sets.

## Imaging excitatory and VIP inhibitory cell populations during task performance

We imaged activity in transgenic mice expressing the calcium indicator GCaMP6f in excitatory pyramidal cells (Slc17a7-IRES2-Cre; CaMKII-tTA; Ai93-GCaMP6f) or VIP inhibitory cells (VIP-IRES-Cre; Ai148-GCaMP6f) (see *Table 1* for numbers of mice, sessions, and cells in the dataset). On average we imaged 181 ± 77 (mean ± SD) Slc17a7+ cells or 15 ± 10 VIP+ cells per session. Measurements were made in primary visual cortex (VISp) and one higher visual area (VISal) but we did not observe major differences between these two areas, so the datasets were combined for the analyses reported here. Because calcium signals have a slow decay time that could lead to an artificial enhancement of the response to stimuli shown close together in time, we performed event detection to identify the onset timing of 'spike' events underlying the GCaMP signal (*de Vries et al., 2020*; *Jewell and Witten, 2017*; *Jewell et al., 2019*). This method produces a timeseries of detected events which have a magnitude proportional to the change in calcium activity. Event magnitude (arbitrary units) was used for all subsequent analysis.

Excitatory cells typically responded to only one or a few images in each set, showing fluorescence increases after stimulus onset or sometimes after stimulus offset (*Figure 2A,B*). VIP cells were less image selective and showed correlated fluctuations in activity (*Figure 2C,D*). Interestingly, VIP cells had distinct activity dynamics relative to stimulus onset in novel versus familiar image sessions. Novel images generated stimulus-locked activity in VIP cells (*Figure 2D*), but this was reduced or absent with familiar images. Instead, during familiar image sessions many VIP neurons had ramping activity that preceded stimulus presentation and decayed after stimulus onset (*Figure 2C*). These ramping responses were even more pronounced when an image presentation was omitted (*Figure 2C*, right panel). In contrast, VIP cells showed little activity during the omission of novel image presentations (*Figure 2D*, right panel).

These differences in image responsiveness and temporal dynamics, already evident in single cell activity, are further quantified across the population of recorded neurons in the subsequent sections.

## Reduced image-evoked activity with familiar image sets

We used a heatmap to visualize activity of the full population of recorded neurons to the eight stimuli for each image set, as well as for omitted stimuli (*Figure 3A*). Most excitatory neurons responded to one of the eight images from a given set and showed little activity when stimuli were omitted. VIP neurons could also show robust image responses, particularly for the novel image sets. In contrast, during sessions with familiar image sets, VIP neurons were most strongly active during the extended gray screen period when stimuli were omitted (*Figure 3A*).

Quantifying each cell's mean response to its preferred image in each set revealed that both excitatory and VIP inhibitory populations had reduced activity levels with familiar images (*Figure 3B*, p<0.008 for all comparisons with image set A for both excitatory and VIP inhibitory populations,

**Table 1.** Number of mice, sessions, and cells in dataset.

| cell class | image set | mice | sessions | total cells |
|---|---|---|---|---|
| Excitatory | A | 11 | 13 | 2046 |
| Excitatory | B | 11 | 13 | 2594 |
| Excitatory | C | 11 | 13 | 2172 |
| Excitatory | D | 11 | 12 | 2232 |
| VIP Inhibitory | A | 10 | 13 | 183 |
| VIP Inhibitory | B | 10 | 13 | 209 |
| VIP Inhibitory | C | 10 | 12 | 186 |
| VIP Inhibitory | D | 10 | 12 | 175 |

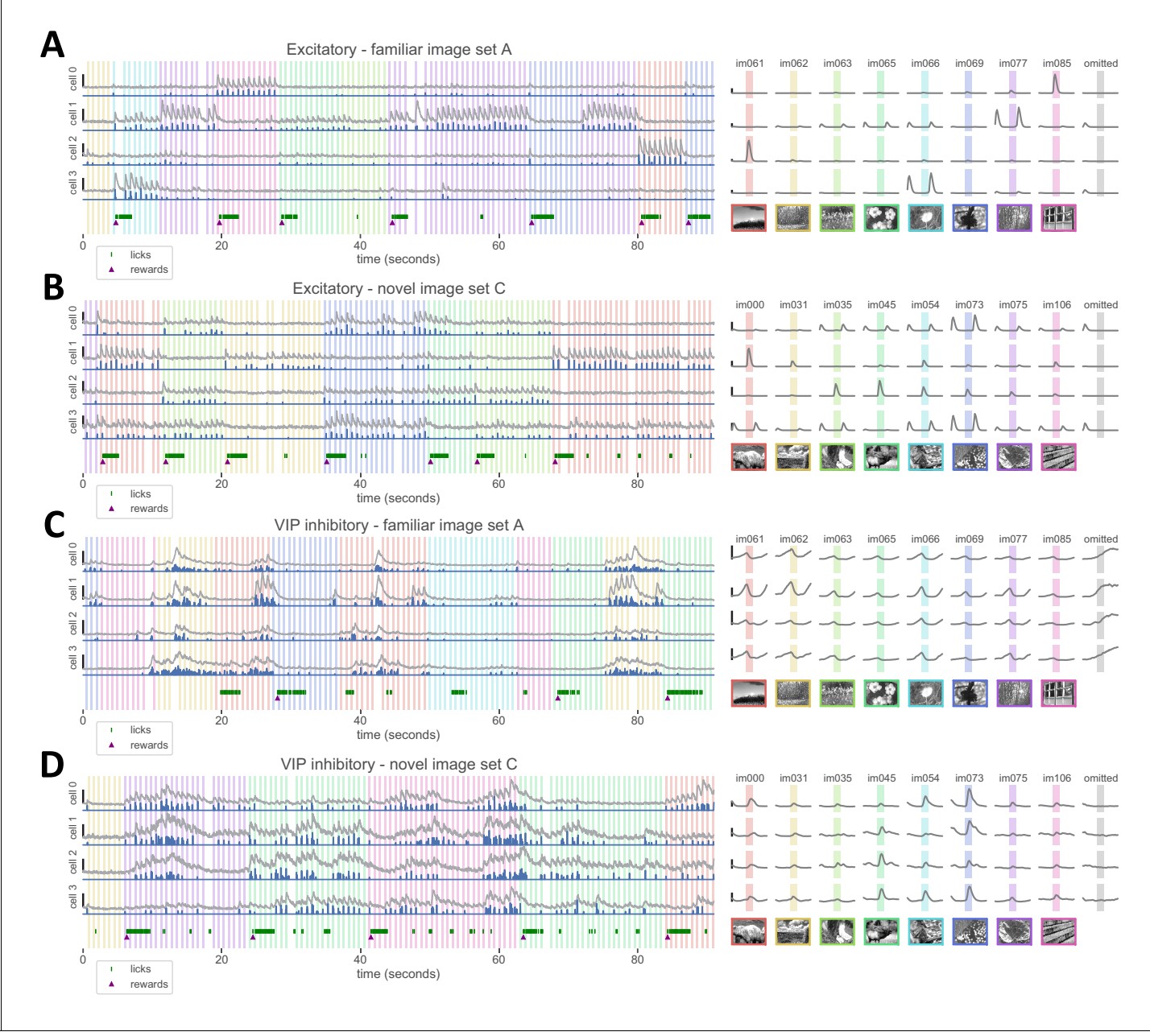

**Figure 2.** Activity in layer 2/3 excitatory and VIP inhibitory cells during change detection task. (**A**) Example cell responses to the familiar image set A, from excitatory cells in layer 2/3 of an Slc17a7-IRES2-Cre;CaMkII-tTa;Ai93 mouse expressing GCaMP6f. Left panel: dF/F traces (gray) from four excitatory cells over a 90 second epoch of a behavior session (scale bars on left indicate 75% dF/F). Deconvolved events are shown in blue below the dF/F trace. Colored vertical bars indicate image presentation times; timing of licks and reward delivery are shown at bottom. Right panel: response of the same 4 cells to each image, as well activity during stimulus omission (right column, gray shading indicates the time where a stimulus would have been displayed). Scale bars indicate 0.05 event magnitude in arbitrary units. (**B**) Example excitatory cells from a session with novel image set C. Left panel scale bars indicate 100% dF/F, right panel scale bars indicate event magnitude of 0.05. (**C**) Example VIP inhibitory cells from layer 2/3 of a VIP-IRES-Cre;Ai148 mouse expressing GCaMP6f, from a session with familiar image set A. Left panel scale bars indicate 225% dF/F, right panel scale bars indicate event magnitude of 0.05. (**D**) Example VIP inhibitory cells for a session with novel image set C. Left panel scale bars indicate 200% dF/F, right panel scale bars indicate event magnitude of 0.05.

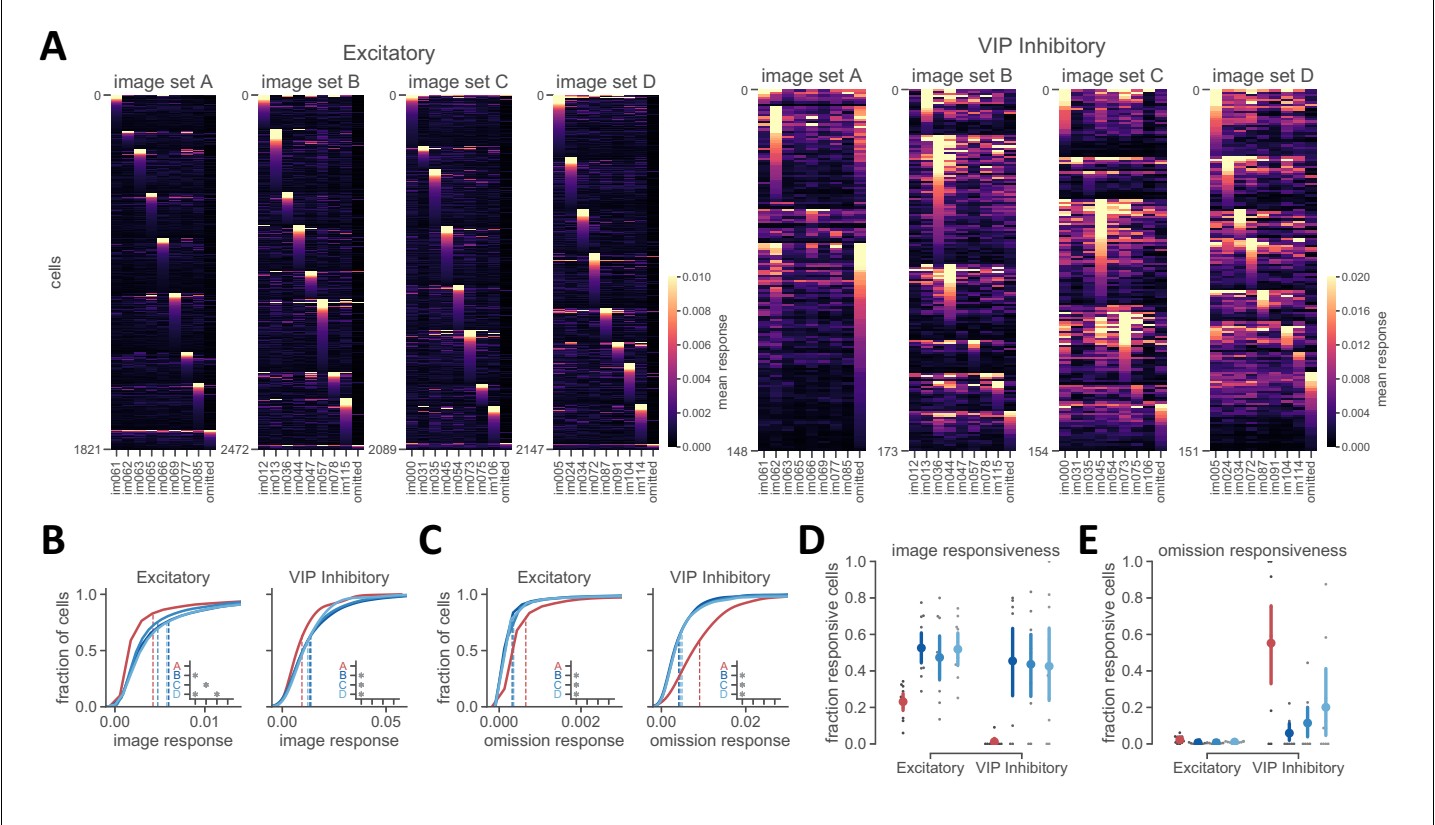

**Figure 3.** Reduced image-evoked activity for familiar stimuli. (**A**) Heatmap showing the mean response of excitatory (left panels) and VIP inhibitory (right panels) cells to images and stimulus omission for familiar and novel image sets. Response is computed using detected events in a 500 ms window after stimulus onset and averaged over all presentations of a given image or image omission. (**B**) Cumulative distribution of response magnitude for each cell's preferred image (excluding omissions) demonstrating reduced image-evoked activity for familiar compared to novel image sets. Insets show comparisons with p<0.008 (Welch's t-test with Bonferroni correction was used for all statistical comparisons, see Materials and methods for additional details). (**C**) Cumulative distribution of omission response magnitude across cells, demonstrating increased activity during stimulus omission for the familiar image set A. Insets are as described in panel **B**. (**D**) Fraction of image responsive cells is higher for novel image sets compared to the familiar image set. Image responsiveness is defined for each cell as having >25% of preferred image stimulus presentations with a significant response compared to a shuffled distribution of values taken from omission periods with extended gray screen. The fraction of image responsive cells is the number of cells within each session that meet the criterion for image responsiveness. Individual sessions are shown in gray, with mean across sessions ± 95% confidence intervals in color. p<0.008 for all comparisons with image set A. (**E**) Fraction of omission responsive cells is higher for the familiar image set in VIP inhibitory cells. Omission responsiveness is defined for each cell as having >10% of stimulus omissions with a significant response compared to a shuffled distribution of values taken from image presentations. The fraction of omission responsive cells is the number of cells within each session that meet the criterion for omission responsiveness. Individual sessions are shown in gray, with mean across sessions ± 95% confidence intervals in color. p<0.008 for A-B and A-C in VIP cells.

The online version of this article includes the following figure supplement(s) for figure 3:

**Figure supplement 1.** Response sparseness for familiar and novel images.

except comparison between A-C for excitatory neurons where p=0.05, Welch's t-test with Bonferroni correction, see Materials and methods for detailed description of statistics used throughout). Both VIP and excitatory cells showed increased stimulus omission activity in familiar image sessions (*Figure 3C*, p<0.008 for all comparisons with image set A). The fraction of cells that were significantly more responsive during image presentations compared to stimulus omission was higher for novel image sets for both cell classes (*Figure 3D*, p<0.008 for all comparisons with image set A). While excitatory cells had a small but significant increase in the fraction of omission responsive cells (2.3% for image set A versus <1% for image sets B-D), a large fraction of VIP cells were omission responsive with the familiar image set compared to novel image sets (55% for image set A, 6–20% for image sets B-D) (*Figure 3E*, for VIP: p<0.008 for A-B and A-C, p=0.03 for A-D; for excitatory:

p=0.01 for A-B and A-C, p=0.05 for A-D). This indicates a trade-off between image responsiveness and omission activity in VIP cells.

We also observed that the responses of individual excitatory neurons to natural images were more selective for familiar compared to novel stimuli. To evaluate single cell image selectivity, we quantified lifetime sparseness (*Vinje and Gallant, 2000*) for image responsive cells. Excitatory populations had higher lifetime sparseness values for the familiar image set compared to the novel image sets (p<0.008 for all comparisons with image set A and for B-C for excitatory cells), and excitatory cells were typically sparser than VIP cells (*Figure 3—figure supplement 1A*). Plotting the population tuning curve for each image set revealed sharper tuning in image responsive excitatory cells for familiar images due to a selective increase in the preferred image response (*Figure 3—figure supplement 1B*), consistent with previous literature (*Woloszyn and Sheinberg, 2012*).Together, these results demonstrate that while overall population activity levels were reduced for familiar images, single cell selectivity was sharpened.

## Inter-stimulus activity dynamics of VIP cells are altered by training history

Next we investigated the temporal dynamics of activity during stimulus presentation and the preceding inter-stimulus interval by examining the average population activity of VIP and excitatory cells for each each image set (*Figure 4A*). The excitatory population showed a sharp increase in activity following stimulus onset, and while familiar images evoked a smaller population response magnitude, the timing of activity was similar across novel and familiar image sets (*Figure 4A*, left panel). In contrast, VIP population dynamics were very different between familiar and novel images sets. With novel images, the VIP population response increased following stimulus onset, but with familiar images, VIP activity ramped up during the inter-stimulus interval and peaked at the time of stimulus onset (*Figure 4A*). This pre-stimulus ramping activity was readily apparent in individual VIP cells but was rare in excitatory cells (*Figure 4B*). Consistent with this effect, the distribution of peak response times across VIP cells was shifted earlier in time for familiar versus novel images sets (*Figure 4C*, p<0.008 for all image set comparisons except B-C for VIP cells). Excitatory cells also showed a small but significant difference in the peak time distribution across image sets (*Figure 4C*, p<0.008 for all comparisons with image set A, as well as B-C for excitatory cells).

To characterize these dynamics across the population, we made use of a ramping index to quantify activity increases or decreases within the pre-stimulus and stimulus epochs. This index compares activity between early and late portions of a defined temporal window and is positive for activity increases and negative for activity decreases (*Makino and Komiyama, 2015*). The distribution of the stimulus ramp index was shifted towards positive values for novel images, consistent with increased image responsiveness (see marginal distributions in *Figure 4D*, p<0.008 for all comparisons with image set A, as well as for B-D and C-D for excitatory cells). The distribution of the pre-stimulus ramp index values was also shifted towards positive values for VIP cells with the familiar image set, indicating an increase in pre-stimulus ramping (see marginals distributions in *Figure 4D*, p<0.008 for all comparisons with image set A, as well as for B-C and C-D for excitatory cells and B-D for VIP cells). The values of the ramp index for the pre-stimulus and stimulus periods were inversely correlated (*Figure 4D*; VIP: r = −0.69, p<0.005; excitatory: r = −0.36, p<0.005; linear least-squares regression fit across all cells). This indicates a tradeoff between stimulus-driven activity and inter-stimulus ramping, particularly in VIP cells.

To further quantify how these response profiles relate to experience, we determined the fraction of cells that were stimulus responsive (positive stimulus ramp index) or stimulus suppressed (negative stimulus ramp index) across image sets (*Figure 4E*). The majority of excitatory neurons were stimulus responsive (<85%), although there was a slight increase in the fraction of stimulus suppressed cells for the familiar image set (1.4% for image set A versus ~0.4% for image sets B-D) (*Figure 4E,F*, left panels; p<0.008 for all comparisons with image set A). VIP inhibitory cells showed a larger difference across image sets: during novel image sessions, most cells were stimulus-driven (<80%), whereas during familiar image sessions, the majority of VIP cells (60%) were stimulus-suppressed and showed pre-stimulus ramping (*Figure 4E,F* right panels; p<0.008 for all comparisons with image set A).

As VIP cell activity has been associated with locomotion and arousal (*Dipoppa et al., 2018*; *Fu et al., 2014*; *Pakan et al., 2016*), we performed several control analyses to determine whether these factors could account for VIP cell pre-stimulus ramping activity. First, we sorted the data

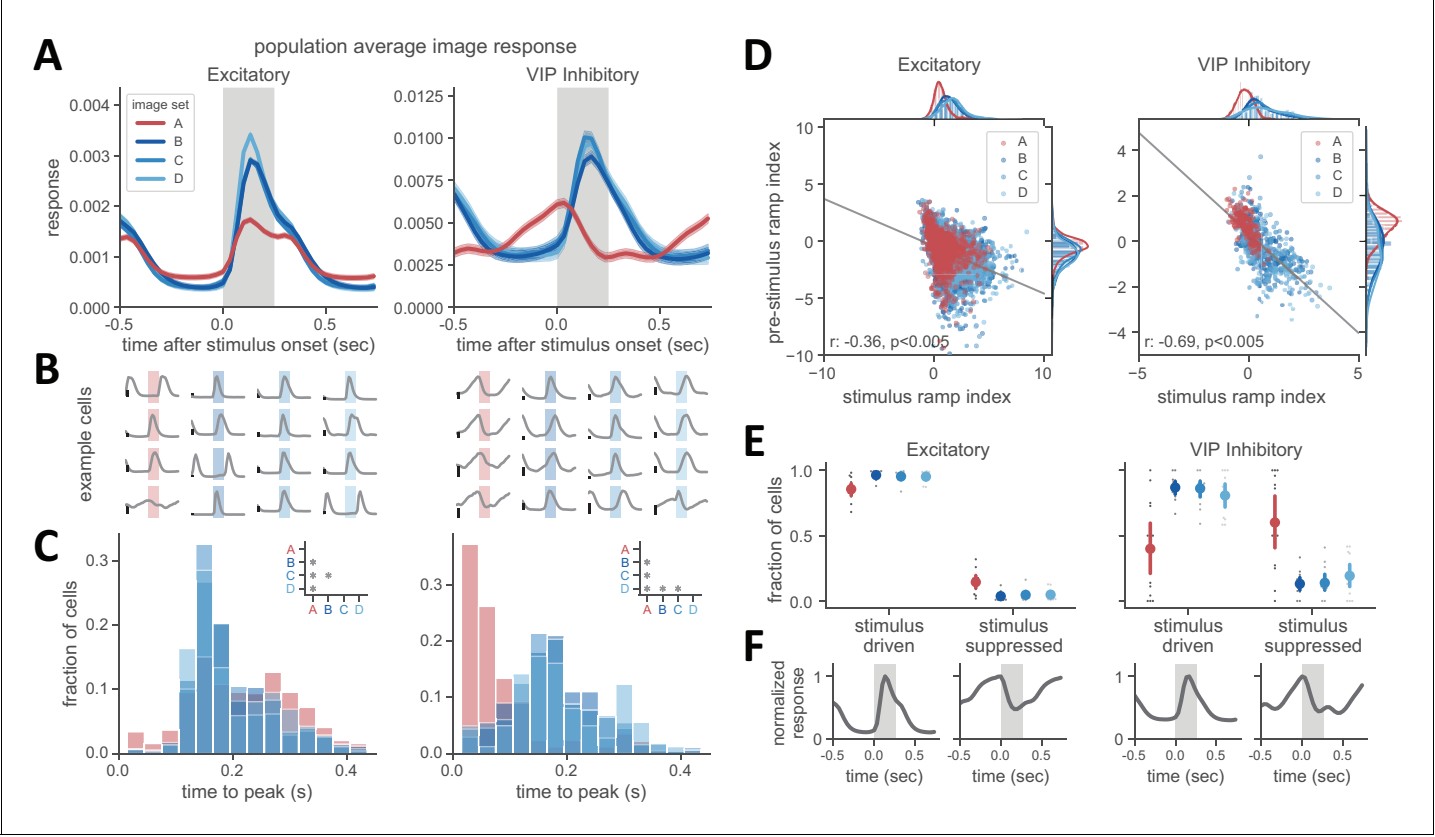

**Figure 4.** Experience-dependent shift in the dynamics of VIP inhibitory cells. (A) Population activity averaged over all image presentations for excitatory (left panel) and VIP inhibitory cells (right panel). Traces show mean ± SEM across cells. Note distinct dynamics in VIP population for novel versus familiar images sets. (B) Example single cell traces showing average image-evoked response for excitatory (left panel) and VIP inhibitory cells (right panel). Background shading denotes stimulus presentation, with color indicating the image set shown during the session for that cell. Each trace represents a unique cell recorded in a single session. Scale bar on left of each trace indicates a response magnitude of 0.005. (C) Histogram of time to peak response after stimulus onset for excitatory (left panel) and VIP inhibitory cells (right panel). Inset shows comparisons across image sets where p<0.008 (Welch's t-test with Bonferroni correction was used for all statistical comparisons, see Materials and methods for additional details). (D) Stimulus ramping and pre-stimulus ramping are negatively correlated. Each point is one neuron. The stimulus ramp index was computed over a 125 ms window after stimulus onset. The pre-stimulus ramp index was computed over a 400 ms window prior to stimulus onset. Data points across all image sets for each panel were fit with linear least-squares regression. Correlation and significance values for the fit are shown in lower left of each panel. (E) Novel image sets have an increased fraction of cells with stimulus-driven activity, whereas a larger fraction of cells was stimulus-suppressed for familiar images. Cells with a positive stimulus ramp index are considered stimulus-driven and those with a negative ramp index are stimulus-suppressed. p<0.008 for all comparisons with image set A. (F) Population average image evoked response for cells that met the criteria for stimulus driven or stimulus-suppressed, as described in panel E.

The online version of this article includes the following figure supplement(s) for figure 4:

**Figure supplement 1.** Changes in image-evoked VIP dynamics are not explained by running behavior or pupil diameter.

according to whether the mouse was running or stationary (see Materials and methods for description of running classification). We found that VIP cells exhibited clear pre-stimulus ramping even when the mouse stationary (*Figure 4—figure supplement 1A–D*). Moreover, we examined stimulus-triggered changes in running speed and found that the pattern of running behavior was similar for novel and familiar sessions, despite a clear difference in VIP cell dynamics (*Figure 4—figure supplement 1C,D*). We also analyzed differences in pupil size across image sets to evaluate whether changes in pupil size around the time of stimulus onset could explain VIP activity dynamics. We found that in novel sessions pupil area was slightly larger on average, but the stimulus-triggered dynamics of pupil area changes were relatively flat and did not match the VIP cell dynamics (*Figure 4—figure supplement 1E,F*). Together these analyses demonstrate that VIP ramping activity does not trivially reflect locomotor or pupil dynamics.

## VIP cells have strong ramping activity during omission of an expected stimulus

Would cells with pre-stimulus ramping activity continue to ramp if an image was omitted? To assess this, we analyzed activity during periods in which stimulus presentations were randomly omitted from the regular sequence. Such trials made up 5% of all non-change stimuli during 2-photon imaging sessions (stimuli were never omitted during behavioral training).

Strikingly, VIP population activity continued to ramp up during stimulus omission, until the subsequent stimulus presentation (*Figure 5A*). Activity following stimulus omission was much stronger during familiar compared to novel image sessions (*Figure 5B*; p<0.008 for all comparisons except A-C for VIP). Omission ramping was not prominent in the excitatory population (*Figure 5A*) but there was a small yet significant increase in the strength of omission activity for the familiar image set (*Figure 5B*, p<0.008 for all comparisons with image set A for excitatory). Visualizing mean omission related activity for all cells as a heatmap for each image set confirmed that very few excitatory cells showed increases in activity following stimulus omission and were primarily active during the stimulus presentations before and after omission (*Figure 5C*). In contrast, most VIP cells showed a dramatic increase in activity following stimulus omission for familiar stimuli (*Figure 5C*). In novel image sessions, VIP cell activity was primarily concentrated outside the omission period, with visible stimulus-locked activity in the surrounding timepoints (*Figure 5C*).

We again assessed whether the dynamics of neural activity were simply correlated with changes in locomotor behavior or pupil area. We found that omission ramping activity was present even when the mice were not running (*Figure 5—figure supplement 1A–D*). Moreover, omission-triggered pupil dynamics were similar during novel and familiar sessions (*Figure 5—figure supplement 1E–F*).

To examine the relationship between omission ramping and pre-stimulus ramping in single cells, we computed the ramp index during the omission window and compared it to the pre-stimulus ramp index. We found a positive correlation between the strength of pre-stimulus and omission ramping for VIP cells (*Figure 5D*; r = 0.49, p<0.005, linear least-squares regression across all VIP cells). While strong omission ramping typically occurred in neurons that had pre-stimulus ramping, it was also possible for cells without pre-stimulus ramping to show increases in activity following stimulus omission (*Figure 5E*). In contrast, pre-stimulus and omission dynamics were not correlated in excitatory cells.

## Discussion

We imaged activity in L2/3 excitatory and VIP inhibitory neurons in response to highly familiar and novel images during a visual task with a predictable temporal structure. This revealed several changes associated with training history. Extended experience with a set of images resulted in reduced overall activity levels in both excitatory and VIP cells. Strikingly, VIP cells exhibited distinct activity dynamics when tested with familiar versus novel images. Novel images drove stimulus-locked activity in VIP cells, whereas with familiar images VIP cells had ramping activity during the inter-stimulus interval and were suppressed by stimulus onset. Moreover, these ramping responses continued increasing when stimuli were omitted from the expected stimulus sequence. The magnitude of VIP omission-related activity was several times larger than stimulus-driven activity, indicating these are meaningful signals that could strongly influence network activity. This represents a major experience-dependent cell type-specific change in response dynamics in sensory cortex.

### Predictive processing and changes in activity with experience

Predictive processing has emerged as a powerful paradigm for understanding brain function and may help reconcile the traditional view of sensory processing with increasing evidence for experience and context-dependent modulation in early sensory areas. This family of theories posits that the brain constructs an internal model of the environment based on experience, and that incoming sensory information is compared with learned expectations to continually update the model (*de Lange et al., 2018*; *Keller and Mrsic-Flogel, 2018*; *Lochmann and Deneve, 2011*; *Rao and Ballard, 1999*). This dynamic updating with experience is proposed to shift the balance of bottom-up sensory and top-down predictive pathways. As stimuli become familiar with learning, predictive signals may

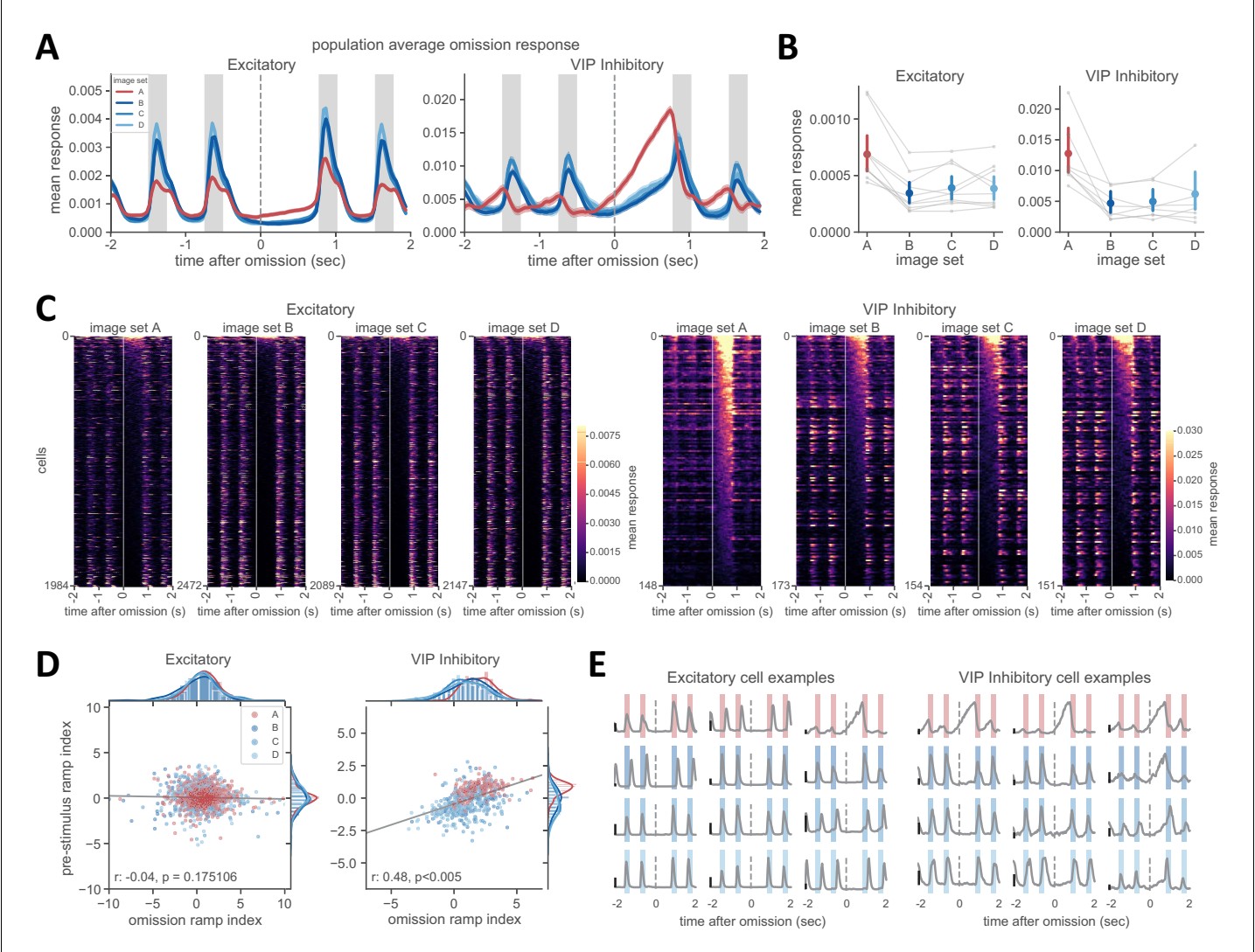

**Figure 5.** VIP cells show strong ramping activity during stimulus omission. (**A**) Average population activity around the time of stimulus omission. On average, excitatory neurons have little change in activity following stimulus omission (left panel). In contrast, activity of the VIP population for the familiar image set A continues to ramp up until the time of the next stimulus presentation. In sessions with novel images (image sets B, C, D), the VIP population also shows some change in activity following stimulus omission, but to a lesser degree than with familiar images. (**B**) Mean activity following stimulus omission is higher during sessions with the familiar image set A. The mean response in a 750 ms window following the time of stimulus omission was first computed for each cell in a given session, then averaged across cells in that session. Connected gray points indicate sessions recorded in a given mouse. Colored points represent the average across sessions for each image set ± 95% confidence intervals. p<0.008 for all comparisons with image set A, except A-C in VIP inhibitory cells (Welch's t-test used for all statistical comparisons, see Materials and methods for additional details). (**C**) Heatmap of activity around the time of stimulus omission across all excitatory (left panels) and VIP inhibitory cells (right panels), sorted by magnitude of activity in the omission window. Start of omission period is shown by white vertical line at time = 0 and extends to 750 ms thereafter when the next stimulus is presented. (**D**) The strength of the omission ramp index (y-axis) and pre-stimulus ramp index (x-axis) are positively correlated across VIP cells, but not excitatory cells, indicating that VIP cells with pre-stimulus activity typically also show ramping during stimulus omission. Data points across all image sets for each panel were fit with linear least-squares regression. Correlation and significance values for the fit are shown in lower left of each panel. (**E**) Example cells showing different response dynamics during stimulus (colored bars) and omission (time of expected stimulus indicated as gray dashed line) for excitatory and VIP inhibitory cells. Color of shaded bars indicates image set (familiar images in red, novel images in blue). Cells typically show either stimulus-evoked activity and no omission response, or pre-stimulus ramping and strong omission responses. Some cells (examples in right column of right panel) show a combination of stimulus-evoked and omission activity. Scale bar indicates a response of 0.01.

The online version of this article includes the following figure supplement(s) for figure 5:

**Figure supplement 1.** VIP omission ramping is not explained by running behavior or pupil diameter.

suppress bottom-up input, resulting in a sparser code. On the other hand, novel or surprising stimuli are expected to robustly drive neural activity, signaling deviations from learned predictions.

We observed reduced activity in both VIP and excitatory cells due to long-term experience with a set of images. The fraction of image responsive cells and mean response magnitude was lower with familiar versus novel images. Previous studies have shown reductions in activity with experience (*Anderson et al., 2008*; *Makino and Komiyama, 2015*; *Mruczek and Sheinberg, 2007*; *Woloszyn and Sheinberg, 2012*), and also novelty enhancement (*Hamm and Yuste, 2016*; *Homann et al., 2017*; *Ranganath and Rainer, 2003*). Reduced activity for highly familiar stimuli can efficiently code predictable stimuli, utilizing a smaller population of cells to represent learned information (*LeMessurier and Feldman, 2018*). On the other hand, enhanced activity for novel stimuli could aid detection of salient and behaviorally meaningful events by augmenting output to downstream targets and facilitating associative plasticity (*Ranganath and Rainer, 2003*).

Our results demonstrating reduced activity with long-term experience and enhancement with novelty are consistent with the predictive coding framework. Increased VIP activity in response to novel images may serve to disinhibit excitatory cell responses to novel stimuli (via the VIP-SST disinhibitory circuit). When stimuli are familiar, VIP cells are suppressed by image presentation, which may permit increased inhibition of excitatory neurons by SST or PV cells, thus increasing the sparseness of stimulus representations. Our observation that VIP cells show ramping activity between familiar images, reminiscent of anticipatory signals, further implicates them in predictive processing.

## Ramping activity in VIP cells

What does pre-stimulus and stimulus-omission ramping activity in VIP cells represent? One possibility is that this activity reflects the temporal structure of the behavioral task such that these signals encode predictions about stimulus timing or reward expectation, or serve as a temporal attention signal (*Nobre and van Ede, 2018*). Previous studies have described stimulus and reward expectation signals in the visual cortex of rodents. An early example showed that pairing visual stimulation with a temporally predictable reward produces reward timing signals in visual cortex (*Shuler and Bear, 2006*). A more recent study using a visual orientation discrimination task in mice found pre-stimulus ramping activity specifically in the subpopulation of excitatory cells that encoded the rewarded stimulus, suggestive of reward anticipation (*Poort et al., 2015*). In our task, each stimulus presentation is a potential opportunity to earn a reward if there is a stimulus change. VIP ramping during the interstimulus interval may be important for enhancing the responses of excitatory cells to an anticipated change stimulus and consequent reward.

Visual cortex has also been shown to learn experience-dependent stimulus predictions for repeated sequences of visual stimuli in the absence of association with reward. One study demonstrated anticipatory recall of an omitted stimulus in a learned spatiotemporal sequence (*Gavornik and Bear, 2014*). In a virtual navigation paradigm in which mice locomote along a linear track, V1 neurons have been found to predict upcoming stimuli at specific locations, and to signal a deviation from expectation when stimuli are omitted (*Fiser et al., 2016*). In these studies, the predictive signal peaked at the expected time of the predicted event, whereas our results show a continued ramping past the expected time of stimulus onset on omission trials. This suggests that the ramping signals we observe may represent something other than a pure prediction of stimulus timing. It is also important to note that most prior studies documenting predictive or ramping activity reflect measurements from excitatory neurons, and thus may not be directly comparable to our results in VIP cells. Nonetheless, VIP cells may serve to gate the predictive or anticipatory signals documented in excitatory cells.

VIP cells in visual cortex have been shown to be modulated by locomotion and arousal in a context-dependent manner (*Dipoppa et al., 2018*; *Fu et al., 2014*; *Pakan et al., 2016*). The ramping activity in VIP cells that we observe, however, is not simply a reflection of locomotion or arousal (as indexed by pupil diameter), since the differences in the dynamics of VIP cells cannot be explained by differences in the pattern of animal running behavior or changes in pupil diameter relative to stimulus and omission onset.

## Role of VIP cells in learning and salience detection

Theoretical work suggests that VIP cells could mediate associative learning and experience-dependent signal routing via disinhibition of local excitatory populations (*Wang and Yang, 2018*; *Wilmes and Clopath, 2018*; *Yang et al., 2016*). Several lines of experimental evidence support this hypothesis. For example, VIP cells in auditory cortex respond to salient reinforcement signals including reward and punishment, and activation of VIP cells enhances the gain of auditory responses (*Letzkus et al., 2011*; *Pi et al., 2013*). Inactivation of VIP cells in visual cortex impairs plasticity following monocular deprivation, while activating VIP cells enhances plasticity (*Fu et al., 2017*). In the amygdala, VIP cells are necessary for associative memory formation through disinhibition of local pyramidal cells during fear learning (*Krabbe et al., 2019*). Further, VIP cells are modulated by expectation (*Krabbe et al., 2019*).

Our finding that VIP cells switch from a stimulus responsive to stimulus suppressed mode depending on past experience with an image set may be consistent with a role for VIP cells in enhancing the representation of salient events to facilitate learning and adaptive behavior. During sessions with novel images, VIP activity is strongly driven by stimulus presentation, potentially serving to increase the gain of stimulus evoked excitatory responses to salient, novel stimuli. When stimuli are familiar, other aspects of the task may become more relevant, such as the unexpected omission of stimuli from a predictable temporal sequence, and VIP cells switch from signaling the presence of a stimulus to signaling the absence of a stimulus.

VIP cells in the visual cortex of passively viewing mice are suppressed by high contrast stimuli and show direction selective responses at low contrast, leading to the proposal that VIP cells are involved in enhancing the gain of weak but salient stimuli (*de Vries et al., 2020*; *Millman et al., 2019*). The inter-stimulus and omission ramping activity we observed, with a decay in the VIP response following onset of familiar images, is consistent with the phenomenon of contrast suppression. However our finding that VIP cells show robust stimulus-evoked responses to novel stimuli demonstrates that VIP responses are flexible and depend on experience. These seemingly divergent findings can be unified under the view that VIP cells represent salient stimuli in a context-dependent manner. Novelty may put cortical circuits in a regime requiring high sensitivity, involving VIP signaling to enhance the gain of salient stimuli. When stimuli are familiar and the environment is more predictable, VIP cells may be more sensitive to unexpected events, such as stimulus omission or violations of temporal expectation. It remains an open question whether VIP activity switches from stimulus driven to stimulus suppressed following passive exposure to a novel set of images in the absence of reward, or whether active task performance and reinforcement are necessary to observe this context-dependent switch in temporal dynamics.

### Future directions

Further modifications of our visual change detection task to include omitted rewards or variation in the predictability of the inter-stimulus interval could help to distinguish between coding of stimulus timing versus reward anticipation. Studies examining the evolution of activity in identified VIP cells across multiple behavior sessions as novel images become familiar are needed to determine the time course of the observed experience dependent effects. Concurrent recordings of VIP, excitatory, and other inhibitory cell classes including SST cells will be important to establish a direct link between VIP activity and disinhibition of local excitatory neurons during task learning. Finally, experiments examining the activity and impact of the diverse inputs to VIP cells, including neuromodulatory inputs (*Lee et al., 2013*; *Letzkus et al., 2011*), thalamic inputs (*Williams and Holtmaat, 2019*), and feedback projections from other cortical regions (*Wall et al., 2016*; *Zhang et al., 2016*; *Zhang et al., 2014*), will be critical to establish the function of and mechanism behind the shift in VIP dynamics with experience.

## Materials and methods

**Key resources table**

| Reagent (species) or resource | Designation | Source or reference | Identifiers | Additional information |
|---|---|---|---|---|

*Continued on next page*

*Continued*

| Reagent (species) or resource | Designation | Source or reference | Identifiers | Additional information |
|---|---|---|---|---|
| Genetic reagent (*M. musculus*) | Slc17a7-IRES2-Cre; Slc17a7+; Excitatory | Jackson Laboratory | Stock #:023527; RRID:Addgene_61574 | PMID: 25741722 |
| Genetic reagent (*M. musculus*) | VIP-IRES-Cre; VIP+; VIP Inhibitory | Jackson Laboratory | Stock #: 010908; RRID:MGI:4436915 | Dr. Z Josh Huang (Cold Spring Harbor Laboratory) |
| Genetic reagent (*M. musculus*) | CaMKII-tTA x Ai93-GCaMP6f | Jackson Laboratory | Stock #: 024108; RRID:IMSR_JAX:024108 | PMID: 22855807; PMID: 25741722 |
| Genetic reagent (*M. musculus*) | Ai148-GCaMP6f | Jackson Laboratory | Stock #: 030328; RRID:IMSR_JAX:030328 | PMID: 30007418 |
| Software, algorithm | numpy | NumPy | RRID:SCR_008633 | |
| Software, algorithm | scipy | SciPy | RRID:SCR_008058 | |
| Software, algorithm | matplotlib | MatPlotLib | RRID:SCR_008624 | |
| Software, algorithm | pandas | pandas | DOI: 10.5281/zenodo.3509134 | |
| Software, algorithm | seaborn | seaborn | DOI: 10.5281/zenodo.1313201 | |

## Mice

All experiments and procedures were performed in accordance with protocols approved by the Allen Institute Animal Care and Use Committee. We used male and female transgenic mice expressing GCaMP6f in VIP inhibitory interneurons (double transgenic: VIP-IRES-Cre x Ai148 mice; https://www.jax.org/strain/010908; https://www.jax.org/strain/030328) (*Daigle et al., 2018*) or in excitatory glutamatergic neurons (triple transgenic: Slc17a7-IRES2-Cre x CaMKII-tTA x Ai93; https://www.jax.org/strain/023527, https://www.jax.org/strain/024108) (*Madisen et al., 2015*; *Mayford et al., 1996*). Mice were single housed and maintained on a reverse 12 hr light cycle (off at 9am, on at 9pm) and all experiments were performed during the dark cycle.

## Surgery

Surgical procedures were performed as described in *de Vries et al. (2020)* (see Supplementary Figure 14). Headpost and cranial window surgery was performed on healthy mice that ranged in age from 5 to 12 weeks. Pre-operative injections of dexamethasone (3.2 mg/kg, S.C.) were administered at 12 hr and 3 hr before surgery. Mice were initially anesthetized with 5% isoflurane (1–3 min) and placed in a stereotaxic frame (Model# 1900, Kopf, Tujunga, CA), and isoflurane levels were maintained at 1.5–2.5% for surgery. An incision was made to remove skin, and the exposed skull was levelled with respect to pitch (bregma-lambda level), roll and yaw. The stereotax was zeroed at lambda using a custom headframe holder equipped with stylus affixed to a clamp-plate. The stylus was then replaced with the headframe to center the headframe well at 2.8 mm lateral and 1.3 mm anterior to lambda. The headframe was affixed to the skull with white Metabond and once dried, the mouse was placed in a custom clamp to position the skull at a rotated angle of 23° such that visual cortex was horizontal to facilitate the craniotomy. A circular piece of skull 5 mm in diameter was removed, and a durotomy was performed. A coverslip stack (two 5 mm and one 7 mm glass coverslip adhered together) was cemented in place with Vetbond (*Goldey et al., 2014*). Metabond cement was applied around the cranial window inside the well to secure the glass window. Post-surgical brain health was documented using a custom photo-documentation system and at one, two, and seven days following surgery, animals were assessed for overall health (bright, alert, and responsive), cranial window clarity, and brain health. After a 1–2 week recovery from surgery, animals underwent intrinsic signal imaging for retinotopic mapping, then entered into behavioral training.

## Intrinsic signal imaging

Intrinsic signal imaging (ISI) was performed as described in *de Vries et al. (2020)* (see Supplementary Figure 15) to produce a retinotopic map to define visual area boundaries and target in vivo two-photon calcium imaging experiments to the center of visual space in each imaged area. Mice were lightly anesthetized with 1–1.4% isoflurane administered with a somnosuite (model #715; Kent Scientific, CON). Vital signs were monitored with a Physiosuite (model # PS-MSTAT-RT; Kent Scientific). Eye drops (Lacri-Lube Lubricant Eye Ointment; Refresh) were applied to maintain hydration and clarity of eye during anesthesia. Mice were headfixed for imaging.

The brain surface was illuminated with two independent LED lights: green (peak λ = 527 nm; FWHM = 50 nm; Cree Inc, C503B-GCN-CY0C0791) and red (peak λ = 635 nm and FWHM of 20 nm; Avago Technologies, HLMP-EG08-Y2000) mounted on the optical lens. A pair of Nikon lenses lens (Nikon Nikkor 105 mm f/2.8, Nikon Nikkor 35 mm f/1.4), provided 3.0x magnification (M = 105/35) onto an Andor Zyla 5.5 10tap sCMOS camera. A bandpass filter (Semrock; FF01-630/92 nm) was used to record reflected red light from the brain.

A 24' monitor was positioned 10 cm from the right eye. The monitor was rotated 30° relative to the animal's dorsoventral axis and tilted 70° off the horizon to ensure that the stimulus was perpendicular to the optic axis of the eye (*Oommen and Stahl, 2008*). The visual stimulus for mapping retinotopy was a 20° x 155° drifting bar containing a checkerboard pattern, with individual square sizes measuring 25°, that alternated black and white as it moved across a mean-luminance gray background. The bar moved in each of the four cardinal directions 10 times. The stimulus was warped spatially so that a spherical representation could be displayed on a flat monitor (*Marshel et al., 2011*).

After defocusing from the surface vasculature (between 500 μm and 1500 μm along the optical axis), up to 10 independent ISI timeseries were acquired and used to measure the hemodynamic response to the visual stimulus. Averaged sign maps were produced from a minimum of 3 timeseries images for a combined minimum of 30 stimulus sweeps in each direction (*Garrett et al., 2014*).

The resulting ISI maps were automatically segmented by comparing the sign, location, size, and spatial relationships of the segmented areas against those compiled in an ISI-derived atlas of visual areas (*de Vries et al., 2020*). A cost function, defined by the discrepancy between the properties of the matched areas, was minimized to identify the best match between visual areas in the experimental sign map and those in the atlas, resulting in an auto-segmented and annotated map for each experiment. Manual correction and editing of the results included merging and splitting of segmented and annotated areas to correct errors. Finally, target maps were created to guide in vivo two-photon imaging location using the retinotopic map. The center of retinotopic space was computed from azimuth and altitude maps and adjusted for variability in eye position relative to the monitor by zeroing to the anatomical center V1. The corresponding retinotopic location was identified for each visual area, and two-photon imaging was targeted to a region within 20° of the center of gaze.

## Behavior training

### Water restriction and habituation

Throughout behavior training mice were water-restricted in order to maintain consistent motivation to learn and perform the behavioral task (*Guo et al., 2014*). Prior to water restriction mice were weighed once daily for three days to obtain a stable, initial baseline weight. During the first week of water restriction mice were handled daily and habituated to increasing duration of head fixation in the behavior enclosure over a five-day period. Thus, the first day of behavior training occurred after 10 days of water restriction. Mice were trained 5 days per week and could earn as much water as possible during the daily one-hour sessions; supplemental water was provided if earned volume fell below 1.0 mL and/or body weight fell under 80–85% of their initial baseline weight. On non-training days mice were weighed and received enough water provision to reach their target weight of 80–85% (never less than 1.0 mL per day).

### Apparatus

Mice were trained in custom-designed, sound-attenuating behavior enclosures. Visual stimuli were displayed on a 24' LCD monitor (ASUS, Model # PA248Q) placed at a ~ 15 cm distance from the

mouse's right eye. The monitor was rotated 30° relative to the animal's dorsoventral axis and tilted 70° off the horizon to ensure that the stimulus was perpendicular to the optic axis of the eye (*Oommen and Stahl, 2008*). A behavior stage was placed in a consistent location using a kinematic mount and consisted of a standardized headframe clamp to enable repeatable positioning of the mouse relative to the monitor, and a 6.5' running wheel tilted upwards by 10–15 degrees (see Supplementary Figure 13 of *de Vries et al., 2020*). Running behavior was measured by a rotational encoder. Water rewards were delivered using a solenoid (NResearch, Model #161K011) that allowed for a calibrated volume of fluid to pass through a blunted, 17 g hypodermic needle (Hamilton) positioned approximately 2–3 mm from the animal's mouth. Licks were detected by a capacitive sensor coupled to the reward delivery spout. Running speed, lick times, and reward delivery times were recorded on a NI PCI-6612 digital IO board and sampled at the frequency of the visual display (60 Hz).

## Behavioral training procedure

Mice were trained for 1 hour/day, 5 days/week using an automated training algorithm. Briefly, mice were trained to lick when the identity of a flashed visual stimulus changed. If mice responded correctly within a short, post-change response window (750 ms) a water reward (5–10 uL) was delivered. On Day 1 of the automated training procedure mice received a short, 15 min 'open loop' conditioning session during which non-contingent water rewards were delivered coincident with 90 degree changes in orientation of a full-field, static square-wave grating. This session was intended to 1) introduce the mouse to the fluid delivery system, 2) provide the technician an opportunity to identify the optimal lick spout position for each mouse and 3) condition the association between stimulus changes and reward delivery. Each session thereafter was run in 'closed loop' mode, and progressed through 3 stages of the operant task (schematized in *Figure 1E*): 1) static, full-field square wave gratings (changes between 0 and 90 degrees, spatial frequency 0.04 cycles per degree), 2) full-field square-wave gratings (changes between 0 and 90 degrees, spatial frequency 0.04 cycles per degree) presented for 250 ms with an 500 ms inter stimulus gray period, and 3) full-field natural scenes (eight natural images from the Allen Brain Observatory) presented for 250 ms with a 500 ms inter stimulus gray period between stimuli. Progression through each stage required mice to achieve a session maximum d-prime of 1 on two of the last three sessions. The shortest amount of time to reach the final stage of training was five sessions. Once in stage 3, mice were considered 'ready for imaging' when 2 out of 3 sequential sessions had a d-prime >1 and mice performed at least 100 trials. However, many mice remained in stage 3 of behavior training until the 2-photon microscope became available. This resulted in a variable training duration in stage three across mice (*Figure 1—figure supplement 1A*).

## Session and trial structure

Each behavior session lasted 60 min and consisted of a continuous series of image presentations with GO and CATCH trials interspersed, schematized in *Figure 1A,B*. Briefly, prior to the start of each trial a change-type and change-time were selected. Change-type was chosen based on predetermined frequencies such that GO and CATCH trials occurred with equal probabilities for sessions with two oriented gratings. For the natural image phase in which there were 64 change-pair possibilities, CATCH frequency was set to 12.5% (1/8 of the number of image transitions). To ensure even sampling of all stimulus transitions, a transition path is selected at random from a matrix of 1000 pre-generated paths. Each path takes a pre-determined route through each of the 64 possible transitions, including same-to-same, or catch, transitions. Once a transition path is completed, another path is chosen at random.

Change times were selected from an exponential distribution ranging from 2.25 to 8.25 s (mean of 4.25 s) following the start of a trial. Catch trial times were drawn from the same distribution such that false alarm rates were measured with the same temporal statistics as change trials, to account for any learning of the temporal distribution of change times. On trials when a mouse licked prior to the change or catch time, the trial was restarted with the same scheduled change or catch time. To prevent mice from getting stuck on a single trial, the number of times a trial could be repeated was limited to five. GO and CATCH trials, when combined with mouse's licking response, yield HIT, MISS, FALSE ALARM, and CORRECT REJECTION trials. In addition to the four trial types described

above, behavior sessions contained a subset of 'free reward' trials (GO trials followed immediately by delivery of a non-contingent reward). Behavior sessions across all phases began with five free-reward trials to help prime engagement with the task. Additionally, to promote continued task engagement, one of these free rewards was delivered after 10 consecutive MISS trials.

Each image was shown an average of 487 times during a given one-hour session. On average, there were 4699 stimulus presentations in each session.

## Two-photon imaging during behavior

### Visual stimulation

Visual stimuli were generated using custom Python scripts written in PsychoPy (https://www.psy-chopy.org/; *Peirce, 2009*; *Peirce, 2007*) and were displayed using an ASUS PA248Q LCD monitor, with 1920 × 1200 pixels. Stimuli were presented monocularly, and the monitor was positioned 15 cm from the mouse's eye and spanned 120° X 95° of visual space. The monitor was rotated 30° relative to the animal's midline and tilted 70° off the horizon to ensure that the stimulus was perpendicular to the optic axis of the eye (*Oommen and Stahl, 2008*).

The monitor was gamma corrected and had a mean luminance of 50 cd/m$^2$. To account for the close viewing angle of the mouse, a spherical warping was applied to all stimuli to ensure that the apparent size, speed, and spatial frequency were constant across the monitor as seen from the mouse's perspective (*Marshel et al., 2011*). Visual stimuli were presented at 60 Hz frame rate.

Visual stimuli consisted of a subset of the natural scene images used in the publicly available Allen Brain Observatory dataset (https://observatory.brain-map.org/visualcoding/; *de Vries et al., 2020*). The 32 natural images that we used originated from three different databases of natural scene images: the Berkeley Segmentation Dataset (images 000, 005, 012, 013, 024, 031, 034, 035, 036, 044, 047, 045, 054, 057) (*Strasburger et al., 2011*), the van Hateren Natural Image Dataset (images 061, 062, 063, 065, 066, 069, 072, 073, 075, 077, 078, 085, 087, 091) (*van Hateren and van der Schaaf, 1998*), and the McGill Calibrated Colour Image Database (images 104, 106, 114, 115) (*Olmos and Kingdom, 2004*). The images were presented in grayscale, contrast normalized, matched to have equal mean luminance, and resized to 1174 × 918 pixels.

### Behavior apparatus

Running speed measurement, lick detection, and reward delivery were performed as described above for behavioral training. The monitor was placed in a fixed location relative to the behavior stage to ensure a consistent relationship between the mouse's eye and the screen. Running speed, lick times, and reward delivery times were recorded on a NI PCI-6612 digital IO board and sampled at the frequency of the visual display (60 Hz).

### Two-photon calcium imaging during behavior

Calcium imaging was performed using a Scientifica Vivoscope (https://www.scientifica.uk.com/prod-ucts/scientifica-vivoscope). Laser excitation was provided by a Ti:Sapphire laser (Chameleon Vision – Coherent) at 910 nm. Pre-compensation was set at −10,000 fs2. Movies were recorded at 30 Hz using resonant scanners over a 400 μm field of view (512 × 512 pixels). Temporal synchronization of calcium imaging, visual stimulation, reward delivery and behavioral output (lick times and running speed) was achieved by recording all experimental clocks on a single NI PCI-6612 digital IO board at 100 kHz.

Behavior sessions under the two-photon microscope were 1 hr in duration, with task parameters identical to stage 3 of the behavior training procedure as described above. In addition, during most two-photon imaging sessions, 5% of stimulus presentations were randomly omitted, excluding the change presentation and the presentation immediately prior to the change. These omitted presentations were added to the experimental protocol partway into the experiment, resulting in 86/101 (85%) imaging sessions including omitted presentations. The 15 sessions without omitted presentations included data from one Slc17a7-IRES2;CaMKII-tTA;Ai93 mouse (four sessions in VISp), and two Vip-IRES-Cre;Ai148 mice (three sessions from VISal, and eight sessions from VISp). Sessions without omitted presentations were excluded from any analysis depending on stimulus omission.

Movies of fluorescence were acquired near the center of retinotopic space in VISp and VISal, using ISI target maps and vasculature images as a guide. Once a cortical region was selected, the

objective was shielded from stray light coming from the stimulus monitor using opaque black tape. All recordings were made at a depth of ~175 um from the brain surface. Once a field of view was selected, a combination of PMT gain and laser power was selected to maximize laser power (based on a look-up table against depth) and dynamic range while avoiding pixel saturation (max number of saturated pixels < 1000). Immersion water was occasionally supplemented while imaging using a micropipette taped to the objective (Microfil MF28G67-5 WPI) and connected to a 5 ml syringe via an extension tubing. At the end of each experimental session, a z-stack of images (± 30 μm around imaging site, 0.1 μm step) was collected to evaluate cortical anatomy and evaluate z-drift during experiment. Experiments with z-drift above 10 μm over the course of the entire session were excluded.

For each field of view, imaging and behavior sessions were conducted using each of the four image sets shown in *Figure 1F*, including the familiar image set A used during behavior training, and three novel image sets first experienced by the mouse during the imaging phase of the experiment. On subsequent imaging days for a given field of view, we returned to the same location by matching (1) the pattern of vessels in epi-fluorescence with (2) the pattern of vessels in two photon imaging and (3) the pattern of cellular labelling in two photon imaging at the previously recorded location. Typically, only one field of view was imaged per mouse, however in 3 out of the 21 mice, fields of view were recorded in both VISp and VISal. In cases where an imaging session failed our QC criteria (for example for z-drift >10 um), the session was retaken. As a result, some sessions with 'novel' image sets B, C or D were the second or third exposure (67% were first exposure, 27% were the second exposure, 6% were the third or fourth exposure). In contrast, mice were exposed to familiar image set A for an average of 17 ± 14 sessions during training.

## Pupillometry

Pupil tracking was performed under 850 nm infrared illumination (OSRAM SYLVANIA Inc, LZ1-10R602-0000 mounted to a Thorlabs LB1092-B-ML bi-convex lens) using a 30 Hz infrared sensitive camera (Allied Vision Technologies Mako G-032B) mated to a 0.73x, 130 mm WD lens (Infinity Infin-iStix, part #213073) and a 845–855 nm bandpass filter (Semrock FF01-850/10-25). The camera and IR LED were mounted to the left of the animal and focused on a short-pass dichroic mirror (Semrock FF750-SDi02−25 × 66, cutoff frequency = 750 nm) positioned between the animal and the monitor, thus allowing tracking of the right (monitor facing) eye and pupil. Pupil diameter was extracted from raw video frames using a processing pipeline based on the DeepLabCut tracking algorithm (*Mathis et al., 2018*). Briefly, a model was fit using hand-annotated sample frames (12 points each from the perimeter of the pupil and eyelid) from multiple imaging rigs, subjects, and lighting conditions. The model was then applied to each frame of the 101 eye tracking videos acquired during imaging sessions, excluding 38 sessions for which frame timing information was incomplete. Points with a minimum confidence value of 0.8 (output by the DeepLabCut model) were used to fit separate ellipses to the eyelid and pupil (*Halir and Flusser, 1998*). Any frames with fewer than six high-confidence eye or pupil tracking points, which generally occurred during blinking/squinting, were not fit (replaced with NaN). Pupil area was then calculated as the area of a circle with a radius equal to the major axis of the ellipse fit. Frames with calculated areas greater than three standard deviations from the mean were excluded, as were the two frames immediately before and after any missing fits. Pupil area was interpolated across periods with missing fits.

## Quality control

All data streams were required to pass strict quality control criteria (*de Vries et al., 2020*; see Supplementary Figure 16). For example, Z-drift of the 2-photon imaging plane over the 1 hr imaging session was quantified by performing phase correlation between the frames of a 100 um z-stack taken after the imaging session and a 500 frame average from the beginning of the 2-photon movie and a 500 frame average at the end of the movie. If the distance between the z-stack frames found to be most correlated with the beginning and end of the movie is greater than 10 um, the session failed QC and was retaken. Only imaging sessions passing all QC criteria were included in this study.

## Data processing

Post-processing of 2-photon imaging data was performed as described in *de Vries et al. (2020)* (see Supplementary Figures 19, 20, 22, 23). For each two-photon imaging session, the image processing pipeline included the following steps: (1) motion correction, (2) image normalization to minimize confounding random variations between sessions, (3) segmentation of ROIs, and (4) ROI filtering. Motion correction was performed using phase correlation and rigid translation. Segmentation was performed by morphological filtering on normalized periodic average images constructed from 400 frame blocks, followed by unification of masks across all blocks. ROI filtering was performed to remove segmented regions that were unlikely to correspond to cell somas, based on attributes including size and shape (for example, small ROIs likely corresponding to apical dendrites were removed).

Following identification of cell ROIs, the following steps were performed to obtain $dF/F$ traces and deconvolved event traces: (1) neuropil subtraction, (2) trace demixing, (3) $dF/F$ computation, (4) L0-regularized event detection. For each ROI, a neuropil mask was created, consisting of a 13 pixel ring around the cell soma, excluding any other ROIs. The raw fluorescence trace was generated by averaging all pixels within each cell ROI and the neuropil mask. A neuropil contamination ratio was computed for each ROI and the calcium trace was modeled as $F_M = F_C + rF_N$, where $F_M$ is the measured fluorescence trace, $F_C$ is the unknown true ROI fluorescence trace, $F_N$ is the fluorescence of the surrounding neuropil, and $r$ is the contamination ratio. After determination of $r$, we computed the true trace as $F_C = F_M - rF_N$, which is used in all subsequent analysis. To avoid artificially correlating neurons' activity by averaging fluorescence over two spatially overlapping ROIs, we demixed the activity of all recorded ROIs, as described in *de Vries et al. (2020)*. A global dF/F trace for each cell was computed, with the baseline $F_0$ determined by a rolling mode of 180 seconds across the raw fluorescence trace. An L0-penalized event detection algorithm was applied to the dF/F trace to obtain a timeseries of calcium events with a magnitude proportional to the increase in calcium activity (https://github.com/jewellsean/FastLZeroSpikeInference; *Jewell and Witten, 2018*; *Jewell and Witten, 2017*). Parameters used for event detection can be found at: https://github.com/matchings/visual_coding_2p_analysis/blob/master/visual_coding_2p_analysis/l0_analysis.py (*Garrett, 2020*; copy archived at https://github.com/elifesciences-publications/visual_coding_2p_analysis). Event timeseries were smoothed with a casual half-Gaussian filter with a standard deviation of 0.065 sec. Temporal alignment was performed to link two-photon acquisition frames (30Hz frame rate) with visual stimulation frames (60Hz frame rate) and associated behavioral signals (licking, running speed, reward delivery, sampled at 60Hz frame rate of visual stimulus). The visual stimulus time nearest to each 2-photon frame time was computed, with the condition that the visual stimulus time must be before the 2-photon acquisition time, to ensure that dF/F responses were not attributed to stimulus or behavior events occurring after the change in the calcium signal.

## Data analysis

All data analysis was performed using custom scripts in Python and relied heavily on pandas (https://pandas.pydata.org/), numpy (https://numpy.org/), and scipy (https://www.scipy.org/). Data visualization was performed using matplotlib (https://matplotlib.org/) and seaborn (https://seaborn.pydata.org/).

### Behavior

Response rates for GO and CATCH trials were calculated by evaluating the fraction of trials of each type where a lick was registered within the 750 ms response window following the change or sham change time (*Figure 1I*, *Figure 1—figure supplement 1E*). The fraction of GO trials with a response is the hit rate and the fraction of CATCH trials with a lick response is the false alarm rate. Response rate was similarly computed for all non-change stimulus presentations, as well as following stimulus omission and for the stimulus presentation directly following stimulus omission (*Figure 1—figure supplement 1E*). The d-prime value for each session (*Figure 1—figure supplement 1D*) was computed as:

$$d' = Z[hit\,rate] - Z[false\,alarm\,rate]$$

Where Z is the inverse of the cumulative distribution function (using scipy.stats.norm.ppf).

Reaction time was calculated as the time to first lick after the start of the change time on GO trials (*Figure 1J*) and displayed using seaborn pointplot. Mean run speed was calculated by taking the average of the running speed trace in a ± 2 s window around the image change time for each GO trial, then averaging across all GO trials in each session (*Figure 1—figure supplement 1B*). The average running speed trace across sessions (*Figure 1—figure supplement 1C*) was computed by averaging the running speed trace across all GO trials in a [−2,6] second window around the change time for GO trials. A histogram of lick times relative to stimulus omission was generated across all 2-photon sessions for all mice. The kernel density estimate of lick times was produced for each image set individually (*Figure 1—figure supplement 1F*) using seaborn kdeplot.

Calculation of all behavior metrics was limited to the portion of the session where the mouse was actively engaged in the behavioral task, where engagement was defined those periods during which the mouse earned at least two rewards per minute. Mice performed 248 engaged GO trials per session on average (range = 83–335).

## Physiology

All analysis was performed on detected events, with arbitrary units designated as 'response' throughout the text and figures.

Neural responses were analyzed with respect to stimulus onset and the time of stimulus omissions. Average stimulus evoked traces were generated by averaging all stimulus presentations for each image (*Figure 2*, right panels) or across all stimulus presentations of all images (*Figure 4A,B*, *Figure 4—figure supplement 1*). Average traces around the time of stimulus omission were generated relative to the time when a stimulus would have been presented (*Figure 2*, right columns of right panels, *Figure 5A,C,D*, *Figure 5—figure supplement 1*). For each individual image presentation (or stimulus omission), the mean response in a 500 ms window after stimulus (or omission) onset was computed (including the 250 ms stimulus duration and 250 ms after to include cells with delayed responses or off responses after stimulus offset). Then the average across all stimulus presentations for each image (or omission) was determined (*Figure 3A*). The cumulative distribution of mean image response magnitude across cells was generated using seaborn distplot, for the preferred image for each cell (*Figure 3B*). The preferred image was identified as the image evoking the largest mean response for each cell. The cumulative distribution of mean omission response magnitude across cells was similarly generated using seaborn distplot (*Figure 3C*).

Image responsiveness (*Figure 3D*) was calculated by first comparing the mean response to each individual stimulus presentation for a cell's preferred image to a shuffled distribution of omission responses from that same cell. Specifically, we resampled the responses to omitted stimuli by drawing randomly with replacement 10,000 times, and then assigned a p-value to each individual stimulus presentation equal to the proportion of resampled omitted-stimulus responses that were larger than the mean response to that stimulus presentation. Neurons were classified as 'image responsive' if at least 25% of presentations of the preferred stimulus had a p-value (with respect to omission responses) less than 0.05. Similarly, omission responsiveness (*Figure 3F*) was calculated by comparing each omission response to a shuffled distribution of image responses from the same cell to get a p-value, then classifying cells as 'omission responsive' if at least 10% of omitted-stimulus responses had a p-value less than 0.05.

To quantify selectivity for individual cells, we used a lifetime sparseness metric, computed using the definition in *Vinje and Gallant (2000)*:

$$Sparseness = \frac{1 - \frac{1}{N}\frac{\left(\sum_i r_i\right)^2}{\sum_i r_i^2}}{1 - \frac{1}{N}}$$

where $N$ is the number of images and $r_i$ is the response of the neuron to image i averaged across trials. Lifetime sparseness was only computed for cells that met the image responsiveness criteria described above. The cumulative distribution of lifetime sparseness values for image responsive cells was generated using seaborn distplot (*Figure 3—figure supplement 1A*). Since there were an insufficient number of image responsive cells for image set A (<10), the distribution of lifetime sparseness

values for this condition was not shown. We created population tuning curves across image responsive cells by rank sorting the mean response to the 8 images shown in each session for each cell, then averaging across cells (*Figure 3—figure supplement 1B*). An insufficient number of VIP cells (<10) were image responsive for image set A, thus a tuning curve was not included for this condition in *Figure 3—figure supplement 1B*.

The population average response was computed by first taking the average stimulus triggered response across all images for each cell (examples in *Figure 4B*), then averaging across cells (*Figure 4A*). The time to peak response was identified for each cell as the time where the average stimulus triggered response in a 500 ms window after stimulus onset reached its maximum value (*Figure 4C*).

The dynamics of cell responses were evaluated by computing a ramp index over different time windows of interest, similar to *Makino and Komiyama (2015)*:

$$Ramp\,index = \log_2\left(R_{late} - R_{early}\right)$$

Where $R_{late)}$ is the mean response in the first half of a defined window of time, and $R_{early)}$ is the second half of the window. This index provides a measure of the magnitude and direction of a change in a signal within the window. For *Figure 4D and E*, the ramp index was computed for two windows: the pre-stimulus window (400 ms prior to stimulus onset, comparing the first 120 ms with the last 120 ms) and the stimulus window (125 ms after stimulus offset, comparing the first 65 ms with the last 65 ms in the window) for the mean events trace for each cell across all stimulus presentations of all images. If the cell trace is increasing during the window, the ramp index is positive. If the cell trace decreasing during the window, the ramp index is negative.

The pre-stimulus and stimulus ramp indices were plotted against each other on a cell by cell basis (*Figure 4D*) and found to be correlated by least squares linear regression between the two measures (using scipy.stats.linregress). Cells with positive values of the stimulus ramp index were considered to be 'stimulus driven' and cells with negative values of the stimulus ramp index were considered to be 'stimulus suppressed' (*Figure 4E,F*). The fraction of cells that fell in each of these categories was calculated for each session, then averaged across sessions for each image set (*Figure 4E*). The population average image response was created by averaging across all cells in each category, regardless of image set (*Figure 4F*).

The population average image response was also computed separately for image presentations where mice were running versus stationary (*Figure 4—figure supplement 1A,B*). Image presentations were classified as running if the mean running speed during the [−0.5, 0.75] second window around stimulus onset was >5 cm/s and as stationary if the mean running speed was <5 cm/s. To confirm this classification, and to evaluate any differences in locomotion and arousal across image sets, we also generated plots of average image triggered running speed and pupil area for stimulus presentations classified as running and stationary (*Figure 4—figure supplement 1C–F*). For both running speed and pupil area, traces aligned to the onset of stimulus presentations were first averaged within each session, then averaged across all sessions for each image set.

For *Figure 4D and E*, the ramp index was computed for two windows: the pre-stimulus window (400 ms prior to stimulus onset, comparing the first 120 ms with the last 120 ms) and the stimulus window (125 ms after stimulus offset, comparing the first 65 ms with the last 65 ms in the window) for the mean events trace for each cell across all stimulus presentations of all images. If the cell trace is increasing during the window, the ramp index is positive. If the cell trace decreasing during the window, the ramp index is negative.

The population average omission response was also computed separately for omissions where mice were running versus stationary (*Figure 5—figure supplement 1A,B*). Image presentations were classified as running if the mean running speed in a ± 2 s window around the time of omission >5 cm/s, and as stationary if the mean running speed in the same window was <5 cm/s. To confirm this classification, and to evaluate any differences in locomotion and arousal across image sets, we also generated plots of average omission triggered running speed and pupil area for omissions classified as running and stationary (*Figure 5—figure supplement 1C–F*). For both running speed and pupil area, traces aligned to the time where a stimulus would have been presented, first averaged within each session, then averaged across all sessions for each image set.

## Statistics

For all statistical comparisons, ANOVA (scipy.stats.f_oneway) was used to test for an overall effect of image set within the excitatory or VIP inhibitory groups, followed by Welch's two-sample t-test (scipy.stats.ttest_ind) for each image set pair, using Bonferroni correction for multiple comparisons to set significance level. p-values are reported throughout the text and figure legends, and significance of comparisons where $p<0.0083$ (an alpha value of 0.05 divided by the number of pairwise comparisons) are indicated by asterisks in figure insets.

## Acknowledgements

We thank Jerome Lecoq and Kevin Takasaki for technical help with the 2-photon microscope, Derric Williams for help with behavior and stimulus control software, Douglas Kim, Janelia Research Campus, Howard Hughes Medical Institute, for providing GCaMP6f, and Saskia de Vries, Brian Hu, and Christof Koch for comments on the manuscript. The authors thank the Allen Institute founder, Paul G Allen, for his vision, encouragement, and support

## Additional information

### Funding

| Funder | Author |
| --- | --- |
| Allen Institute for Brain Science | Marina Garrett<br>Sahar Manavi<br>Kate Roll<br>Douglas R Ollerenshaw<br>Peter A Groblewski<br>Justin T Kiggins<br>Linzy Casal<br>Kyla Mace<br>Ali Williford<br>Arielle Leon<br>Xiaoxuan Jia<br>Stefan Mihalas<br>Nicholas D Ponvert<br>Peter Ledochowitsch<br>Michael A Buice<br>Wayne Wakeman<br>Shawn R Olsen |

The funders had no role in study design, data collection and interpretation, or the decision to submit the work for publication.

### Author contributions

Marina Garrett, Conceptualization, Data curation, Software, Formal analysis, Supervision, Validation, Investigation, Visualization, Methodology, Writing - original draft, Project administration; Sahar Manavi, Data curation, Software, Formal analysis, Investigation, Visualization, Methodology; Kate Roll, Data curation, Validation, Investigation, Methodology; Douglas R Ollerenshaw, Conceptualization, Software, Methodology, Writing - review and editing; Peter A Groblewski, Conceptualization, Supervision, Methodology, Project administration, Writing - review and editing; Nicholas D Ponvert, Software, Formal analysis, Writing - review and editing; Justin T Kiggins, Conceptualization, Software, Methodology; Linzy Casal, Ali Williford, Project administration; Kyla Mace, Investigation; Arielle Leon, Methodology; Xiaoxuan Jia, Software; Peter Ledochowitsch, Michael A Buice, Software, Validation; Wayne Wakeman, Software, Project administration; Stefan Mihalas, Conceptualization, Methodology; Shawn R Olsen, Conceptualization, Supervision, Methodology, Writing - original draft, Project administration

### Author ORCIDs

Marina Garrett (iD) https://orcid.org/0000-0002-5271-2291
Peter A Groblewski (iD) http://orcid.org/0000-0002-8415-1118

Stefan Mihalas (iD) http://orcid.org/0000-0002-2629-7100
Shawn R Olsen (iD) https://orcid.org/0000-0002-9568-7057

## Ethics

Animal experimentation: All experiments and procedures were performed in accordance with protocols (#1801) approved by the Allen Institute Animal Care and Use Committee (IACUC).

## Decision letter and Author response

Decision letter https://doi.org/10.7554/eLife.50340.sa1
Author response https://doi.org/10.7554/eLife.50340.sa2

## Additional files

### Supplementary files
• Transparent reporting form

### Data availability

Figshare DOI: https://doi.org/10.6084/m9.figshare.c.4858779.v1.

The following dataset was generated:

| Author(s) | Year | Dataset title | Dataset URL | Database and Identifier |
|---|---|---|---|---|
| Garrett M, Manavi S, Roll KR, Ollerenshaw D, Groblewski P, Ponvert ND, Kiggins JT, Casal L, Mace K, Williford A, Leon A, Jia X, Ledochowitsch P, Buice MA, Wakeman W, Mihalas S, Olsen SR | 2020 | Experience shapes activity dynamics and stimulus coding of VIP inhibitory cells | https://doi.org/10.6084/m9.figshare.c.4858779.v1 | figshare, 10.6084/m9.figshare.c.4858779.v1 |

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
