## [Decision Letter]

**Acceptance summary:**

This study demonstrates for the first time that VIP interneurons in the mouse visual cortex display a learnt response to an expected stimulus (overstrained image repetition), which takes the form of a temporal ramping of activity before the expected stimulus. This ramping is prolonged when the expected stimulus is omitted. These results indicate that VIP cells mediate some kind of predictive information within the circuit of visual cortex.

**Decision letter after peer review:**

Thank you for submitting your article "Experience shapes activity dynamics and stimulus coding of VIP inhibitory and excitatory cells in visual cortex" for consideration by *eLife*. Your article has been reviewed by three peer reviewers, including Brice Bathellier as the Reviewing Editor and Reviewer #1, and the evaluation has been overseen by a Reviewing Editor and Joshua Gold as the Senior Editor. The following individuals involved in review of your submission have agreed to reveal their identity: Georg B Keller (Reviewer #2).

The reviewers have discussed the reviews with one another and the Reviewing Editor has drafted this decision to help you prepare a revised submission.

Summary:

In this work Garrett and colleagues investigate the effect of experience on the response of excitatory and VIP positive inhibitory neurons to repeated stimulus presentations. The key finding is that VIP interneurons exhibit anticipatory responses when the mouse has extensive experience with an image set. These anticipatory responses appear to be suppressed by the presentation of the expected image as they continue increasing when the image is omitted. In addition, the study shows that V1 neurons tend to decrease responsiveness to frequent stimuli as compared to more novel stimuli, a result that is expected and not novel.

The study by Garret et al., is elegantly designed, rigorous and compelling. The key observations are interesting and topical as not much is known about how experience (in this case visual experience) influences VIP activity and as they provide further evidence for the implementation of predictive-like codes in visual cortex. The paper is overall well written and clear, but would benefit from streamlining, to read less as a collection of results.

In addition, a number of points need to be addressed to strengthen the observations and their interpretation.

Essential revisions:

1) The paper would profit from streamlining and condensing. Currently, it reads a bit like a random collection of findings of unclear relation. The authors should focus the paper on their main finding of ramping (and omission) responses.

2) "change modulation index" – Figure 4D. The authors' analysis would suggest that in VIP neurons the response to the 10th presentation is larger than the response to the 1st presentation. Given Figure 4C (right panel), potentially indicating that the stimulus driven response in VIP is also larger for the 1st presentation. The 10th presentation is simply riding on top of a slow decay (and hence appears larger when not baseline subtracted – the measure to quantify response to a stimulus using calcium imaging is stimulus-induced change in dF/F). This slow decay could be the result of sustained spiking activity, or a slower calcium decay kinetics. Unless the authors want to calibrate calcium decay differences between excitatory and VIP neurons, or redo all the experiments with inter-stimulus intervals that would allow for a calcium decay to baseline, they should rework or actually remove this analysis – the most likely explanation for this is a slower calcium decay.

3) The ramping effect in VIP neurons is observed in via calcium signals, which have a relatively slow time constant Tau (even if GCAMP6f is used) and which are not temporally deconvolved. Within a timescale of a few Tau's, constant firing will lead to a ramping of the calcium profile. It would be interesting that the authors either provide deconvolved versions of the observed ramping profile or at least simulations of the calcium signals for constant firing, given the range of time constants observed for GCAMP6f in pyramidal cell and VIP neurons. Clearly, the time constants are dependent on cell types, so it would be good to discuss to which extent ramping response could be confused with (nonetheless quite interesting) steady responses.

4) Although the behavior in which the animal is involved is well controlled, the authors do not provide data on saccades or pupil dilation. Could adaptation to known stimuli be related to pupil size / attentional processes? Is there a change of pupil size with novel stimuli? Or are there saccades related to stimulus change that could contribute to the larger responses? Ideally this should be checked with pupil tracking or at least discussed.

5) It is also important to sort the analysis based on locomotion state. It has been widely studied that locomotion is highly correlated with the activity of VIP neurons. Furthermore, in this manuscript (as in other studies) VIP cell activity is correlated with each other, and possibly triggered by locomotor activity. The authors mentioned that there is no overall changes in the locomotor activity between familiar, novel or omission trials, however it is possible that locomotor changes at specific periods during the trial impacts the activity of VIP cells and influence the interpretation of the results.

6) The authors provide a measure of reaction time as the time of the first lick after stimulus change. But in a supplementary figure they show that reaction times can also be measured based on running speed. It is important to show how similar are the values and if they both do not depend on stimulus novelty.

---

## [Author Response]

Essential revisions:1) The paper would profit from streamlining and condensing. Currently, it reads a bit like a random collection of findings of unclear relation. The authors should focus the paper on their main finding of ramping (and omission) responses.

We have substantially revised and streamlined the manuscript to focus on our major novel finding of ramping (and omission) activity of VIP cells. We have removed analysis and figure panels that are ancillary to this main focus. Specifically, we removed the previous Figure 4 and associated text that described short-term adaptation of responses to repeated stimulation. We have also removed analysis of “change responses” previously included in Figure 3 and Figure 4 and Supplementary figure 3 and Supplementary figure 4.

By focusing our manuscript, we have condensed the main text from 6558 words to 4603 words. This should make the manuscript and our key results more digestible. Where we added new figures (Figure 4—figure supplement 1 and Figure 5—figure supplement 1), these that provide important control analyses of running and pupil size that strengthen our conclusions as suggested by the reviewers below.

2) "change modulation index" – Figure 4D. The authors' analysis would suggest that in VIP neurons the response to the 10th presentation is larger than the response to the 1st presentation. Given Figure 4C (right panel), potentially indicating that the stimulus driven response in VIP is also larger for the 1st presentation. The 10th presentation is simply riding on top of a slow decay (and hence appears larger when not baseline subtracted – the measure to quantify response to a stimulus using calcium imaging is stimulus-induced change in dF/F). This slow decay could be the result of sustained spiking activity, or a slower calcium decay kinetics. Unless the authors want to calibrate calcium decay differences between excitatory and VIP neurons, or redo all the experiments with inter-stimulus intervals that would allow for a calcium decay to baseline, they should rework or actually remove this analysis – the most likely explanation for this is a slower calcium decay.

Following the reviewers’ suggestion to focus and streamline the manuscript we have removed this analysis of “change modulation” from Figure 4 and the text.

3) The ramping effect in VIP neurons is observed in via calcium signals which have a relatively slow time constant Tau (even if GCAMP6f is used) and which are not temporally deconvolved. Within a timescale of a few Tau's, constant firing will lead to a ramping of the calcium profile. It would be interesting that the authors either provide deconvolved versions of the observed ramping profile or at least simulations of the calcium signals for constant firing, given the range of time constants observed for GCAMP6f in pyramidal cell and VIP neurons. Clearly, the time constants are dependent on cell types, so it would be good to discuss to which extent ramping response could be confused with (nonetheless quite interesting) steady responses.

We have now re-analyzed all of the data in our study using temporally deconvolved calcium signals. We used a published method for L0-regularized event detection to identify ‘events’ from the dF/F traces (Jewell and Witten, 2017). All the major results of our study are unchanged in this analysis. Importantly, the ramping activity of VIP cells during inter-stimulus intervals and during stimulus omission is clearly present in these deconvolved event traces. Since this event-based analysis provides a better estimate of the underlying spike rates compared to dF/F, we have replaced all figures with analysis performed on events.

4) Although the behavior in which the animal is involved is well controlled, the authors do not provide data on saccades or pupil dilation. Could adaptation to known stimuli be related to pupil size / attentional processes? Is there a change of pupil size with novel stimuli? Or are there saccades related to stimulus change that could contribute to the larger responses? Ideally this should be checked with pupil tracking or at least discussed.

We collected pupil tracking data for a subset of our experiments (88 out of 101) and now provide an analysis of these data in the manuscript. We show in Figure 4—figure supplement 1 that pupil size is slightly increased in novel stimulus sessions relative to familiar sessions. In the Results section and Discussion section we describe how this could reflect a difference in arousal/attention and might contribute to the higher stimulus response magnitudes of excitatory and VIP cells in the novel image session (Figure 3). Importantly, however, a simple gain change cannot account for our main finding of a shift in VIP cell temporal dynamics between novel and familiar sessions (Figure 4—figure supplement 1 and Figure 5—figure supplement 1).

5) It is also important to sort the analysis based on locomotion state. It has been widely studied that locomotion is highly correlated with the activity of VIP neurons. Furthermore, in this manuscript (as in other studies) VIP cell activity is correlated with each other, and possibly triggered by locomotor activity. The authors mentioned that there is no overall changes in the locomotor activity between familiar, novel or omission trials, however it is possible that locomotor changes at specific periods during the trial impacts the activity of VIP cells and influence the interpretation of the results.

We have now performed an analysis in which we sort our dataset according to running state (running or stationary). We show in Figure 4—figure supplement 1 and Figure 5—figure supplement 1 that VIP cells exhibit inter-stimulus and omission ramping activity both during running and stationary periods. Thus, this phenomenon of ramping activity in VIP cells does not simply reflect locomotor behavior. Moreover, we examined stimulus-triggered (and omission-triggered) locomotor changes and found these are similar for novel and familiar sessions but VIP cell dynamics are different (Figure 4—figure supplement 1 and Figure 5—figure supplement 1).

6) The authors provide a measure of reaction time as the time of the first lick after stimulus change. But in a supplementary figure they show that reaction times can also be measured based on running speed. It is important to show how similar are the values and if they both do not depend on stimulus novelty.

The reviewer is correct that running speed also provides a measure of reaction time. In Figure 1—figure supplement 1 we now show how changes in running speed correspond to lick reaction time. We find that reaction times based on running are quicker (that is, mice begin to slow before they lick). However, running reaction times are very similar for both novel and familiar images and so this cannot account for differences in VIP cell dynamics.